# HIV-1 DNA predicts disease progression and post-treatment virological control

**James P Williams**[1†]**, Jacob Hurst**[1,2†]**, Wolfgang Stöhr**[3]**, Nicola Robinson**[1,2,13]**, Helen Brown**[1,2,13]**, Martin Fisher**[4]**, Sabine Kinloch**[5]**, David Cooper**[6§]**, Mauro Schechter**[7]**, Giuseppe Tambussi**[8]**, Sarah Fidler**[9]**, Mary Carrington**[10,11]**, Abdel Babiker**[3]**, Jonathan Weber**[9]**, Kersten K Koelsch**[6,12‡]**, Anthony D Kelleher**[6,12‡]**, Rodney E Phillips**[1,2,13‡]**, John Frater**[1,2,13*¶‡]**on behalf of the SPARTAC trial investigators**

[1]Nuffield Department of Clinical Medicine, John Radcliffe Hospital, Oxford, United Kingdom; [2]The Oxford Martin School, Institute for Emerging Infections, Oxford, United Kingdom; [3]Medical Research Council Clinical Trials Unit, Institute of Clinical Trials and Methodology, University College London, London, United Kingdom; [4]Department of Sexual Health and HIV, Brighton and Sussex University Hospitals, Brighton, United Kingdom; [5]Division of Infection and Immunity, School for Life Sciences, University College London, London, United Kingdom; [6]The Kirby Institute of New South Wales, Sydney, Australia; [7]Hospital Escola São Francisco de Assis, Universidade Federal do Rio de Janeiro, Rio de Janeiro, Brazil; [8]Department of Infectious Diseases, Ospedale San Raffaele, Milan, Italy; [9]Division of Medicine, Wright Fleming Institute, Imperial College, London, United Kingdom; [10]Cancer and Inflammation Program, Laboratory of Experimental Immunology, Leidos Biomedical Research Inc., Frederick National Laboratory for Cancer Research, Frederick, United States; [11]Ragon Institute of MGH, MIT and Harvard, Cambridge, United States; [12]St Vincent's Centre for Applied Medical Research, Sydney, Australia; [13]Oxford National Institute of Health Research Biomedical Research Centre, Oxford, United Kingdom

*For correspondence: john.frater@ndm.ox.ac.uk

†These authors contributed equally as co-first authors to this work

‡These authors contributed equally to this work

**Present address:** §St Vincent's Centre for Applied Medical Research, Sydney, Australia; ¶Peter Medawar Building for Pathogen Research, Nuffield Department of Medicine, University of Oxford, Oxford, United Kingdom

**Competing interests:** The authors declare that no competing interests exist.

**Reviewing editor**: Quarraisha Abdool Karim, University of KwaZulu Natal, South Africa

**Abstract** In HIV-1 infection, a population of latently infected cells facilitates viral persistence despite antiretroviral therapy (ART). With the aim of identifying individuals in whom ART might induce a period of viraemic control on stopping therapy, we hypothesised that quantification of the pool of latently infected cells in primary HIV-1 infection (PHI) would predict clinical progression and viral replication following ART. We measured HIV-1 DNA in a highly characterised randomised population of individuals with PHI. We explored associations between HIV-1 DNA and immunological and virological markers of clinical progression, including viral rebound in those interrupting therapy. In multivariable analyses, HIV-1 DNA was more predictive of disease progression than plasma viral load and, at treatment interruption, predicted time to plasma virus rebound. HIV-1 DNA may help identify individuals who could safely interrupt ART in future HIV-1 eradication trials.
Clinical trial registration: ISRCTN76742797 and EudraCT2004-000446-20

## Introduction

Renewed interest in exploring avenues for curing Human Immunodeficiency Virus Type 1 (HIV-1) infection (*Henrich et al., 2013*; *Persaud et al., 2013*; *Saez-Cirion et al., 2013*; *Denton et al., 2014*; *Tebas et al., 2014*) has resulted in the investigation of interventions to eradicate cells in which HIV-1 persists despite antiretroviral therapy (ART). Once HIV-1 has infected a cell and integrated its genome into the

**eLife digest** HIV is a virus that can hide in, and hijack, the cells of the immune system and force them to make new copies of the virus. This eventually destroys the infected cells and weakens the ability of a person with HIV to fight off infections and disease. If diagnosed early and treated, most people with HIV now live long and healthy lives and do not develop AIDS—the last stage of HIV infection when previously harmless, opportunistic infections can become life-threatening. However, there are still numerous hurdles and challenges that must be overcome before a cure for HIV/AIDS can be developed.

Treatment with drugs called antiretrovirals can reduce the amount of the HIV virus circulating in an infected person's bloodstream to undetectable levels. However, when HIV infects a cell, the virus inserts a copy of its genetic material into the cell's DNA—and, for most patients, antiretroviral treatment does not tackle these 'hidden viruses'. As such, and in spite of their side-effects, antiretroviral drugs have to be taken for life in case the hidden viruses re-emerge.

As research into a cure for HIV/AIDS gathers momentum, patients who might be candidates for new experimental treatments will need to be identified. Although it is not recommended as part of standard clinical care, the only way to test if a patient's viral levels would remain suppressed without the drugs would be to temporarily stop the treatment under the close supervision of a physician. As such, a new method is needed to identify if there are patients who might benefit from stopping antiretroviral therapy, and more importantly, those who might not.

Williams, Hurst et al. have now tested whether measuring the levels of HIV DNA directly might help to predict if, and when, the virus might re-emerge (or rebound). In a group of HIV patients participating in a clinical trial, those with higher levels of HIV DNA at the point that the treatment was stopped were found to experience faster viral rebound than those with lower levels of HIV DNA. This method could therefore identify those patients who are at the greatest risk of HIV viral rebound, and are therefore unlikely to benefit if their treatment is interrupted.

Williams, Hurst et al. also found that measuring the levels of HIV DNA could help to predict how the disease would progress in treated and untreated patients. Furthermore, these predictions were more accurate than those based on measuring the amount of the virus circulating in a patient's body.

The next challenge is to identify other methods to distinguish patients who may remain 'virus-free' for a period without treatment, from those who would not. With this achieved, it might be possible to identify the mechanisms that determine why the virus comes back and so develop new treatments to stop this happening. This would make developing a cure for HIV/AIDS a much more tangible prospect.

cellular DNA, that cell may revert to a resting state, only producing replication competent virions when activated at a later date. These cells have been labeled the 'HIV reservoir'. There is, however, a lack of clarity relating to the cell types that might harbour the 'reservoir', as well as the tissues in which these cells might be located. For clarity, we will use the term 'reservoir' to describe the population of HIV-1-infected cells that persist during ART and which are the source of rebound viraemia on stopping therapy. Current understanding is that the majority of cells comprising this reservoir are CD4 T memory cells of a resting phenotype (*Avettand-Fenoel et al., 2009*; *Liszewski et al., 2009*; *Eriksson et al., 2013*).

Many assays have been developed to quantify the HIV-1 reservoir, ranging from simple quantitative PCR (qPCR) estimation of cell associated HIV-1 DNA to labour-intensive viral outgrowth assays (VOA) (*Siliciano and Siliciano, 2005*; *Avettand-Fenoel et al., 2009*; *Liszewski et al., 2009*; *Eriksson et al., 2013*). Whereas measurement of plasma viraemia ('viral load') and CD4 T cell count are documented surrogate markers of HIV clinical progression, the clinical relevance and utility of measuring the reservoir—regardless of assay–remains less clear. As cell-associated HIV-1 DNA precedes plasma viraemia in the viral life cycle, it is tantalizing to speculate whether measuring HIV-1 DNA (as a surrogate for reservoir size) might have significant clinical relevance.

It is well documented that HIV-1 DNA persists in patients on antiretroviral therapy (ART) even when the plasma viral load is undetectable using the most sensitive assays (*Palmer et al., 2008*; *Saez-Cirion et al., 2013*). Much of this detectable HIV-1 DNA has been found to be mutated and replication-incompetent calling into question its biological relevance (*Ho et al., 2013*). However, as a simple

surrogate measure of the reservoir it may still have a role to play. As new interventions to cure HIV-1 infection are developed and taken into clinical trials, a means to measure their efficacy is needed. Stopping ART to await the return of viraemia would be the 'gold standard' approach, but has been associated with risk in certain (*Strategies for Management of Antiretroviral Therapy (SMART) Study Group et al., 2006*), although not all studies (*SPARTAC Trial Investigators et al., 2013*). Ideally, the clinician would have access to an algorithm of biomarker assays to help identify those patients who might (or, alternatively, should not) be candidates for a safe treatment interruption (TI). The best way to assess the patient successfully managed on ART is unclear but, with the viral load rendered undetectable, it is plausible that HIV-1 DNA might be an alternative biomarker for disease progression. For example, compared with individuals with uncontrolled viraemia, HIV-1 DNA levels are much lower in cohorts such as VISCONTI in which apparently persistent aviraemia has been reported following TI (*Saez-Cirion et al., 2013*), and in the case of the Mississippi baby extremely low DNA levels were associated with a prolonged period of virological remission. However, this contrasts with cases in which undetectable DNA on ART was associated with prompt rebound viraemia on stopping (*Chun et al., 2010*; *Henrich et al., 2014*). We therefore wished to gain a broader picture of the utility of measuring HIV-1 DNA levels by studying participants in a large, randomized trial of primary HIV-1 infection.

We measured both Total and Integrated HIV DNA levels in peripheral blood CD4 T cells in participants in the Short Pulse Antiretroviral Treatment at HIV-1 Seroconversion (SPARTAC) trial (*SPARTAC Trial Investigators et al., 2013*)–the largest randomized clinical trial of short-course ART in primary HIV-1 infection (PHI). Studying individuals recruited at PHI, randomized to no treatment or ART, and who subsequently underwent treatment interruption, allowed us to ask two questions. Was HIV-1 DNA independently predictive of clinical progression, and did HIV-1 DNA predict the time taken for viraemic rebound on stopping therapy, advocating its role in future treatment interruption protocols?

## Results

### SPARTAC trial participant characteristics

154 participants across all the SPARTAC trial arms were studied based on infection with subtype B HIV-1 and sample availability. All 154 patients were sampled at the pre-therapy baseline at trial enrolment. The demographics of the 154 participants are shown in *Table 1*. Participants who were randomised to receive no therapy or 48 weeks of ART and for whom samples were available (n = 51 and n = 47, respectively; *Supplementary file 1*) were studied in separate analyses described below. Assays of both Total and Integrated HIV-1 DNA were conducted at pre-therapy 'baseline' (trial week 0) and then at weeks 12, 48, 52, 60 and 108, where samples permitted. As detailed in *Supplementary file 2*, not all patients were assayed at all time-points, dependent on the analyses being conducted and sample availability.

### Pre-ART HIV-1 DNA associates with surrogate markers of disease progression

Traditionally, plasma viral load (VL) (*Mellors et al., 1996*) and CD4 cell count (*Frater et al., 2014*) are the only validated surrogate markers of progression used in the HIV-1 clinic. We therefore measured these biomarkers as well as HIV-1 DNA in 154 SPARTAC participants at enrolment to the trial and prior to any ART being given. The median (interquartile range) values of Total and Integrated HIV-1 DNA values in PHI (*Figure 1—figure supplement 1*) were 7707 (2477–18187) and 3830 (1563–6325) copies of HIV-1 DNA per million CD4 T cells, respectively. Total and Integrated HIV-1 DNA levels were closely associated ($p < 0.0001$; $r^2 = 0.72$; Pearson correlation) (*Figure 1—figure supplement 2*) in these pre-therapy samples. Total and Integrated HIV-1 DNA were significantly associated with plasma viral load (both $p < 0.001$; $r^2 = 0.48$ and $0.64$, respectively; linear regression) (*Figure 1A*), and inversely with CD4 T cell count (both $p < 0.001$; $r^2 = 0.20$ and $0.27$, respectively; linear regression) (*Figure 1B*). Interestingly, the estimated time since seroconversion at recruitment did not correlate with HIV-1 DNA (both Integrated and Total) (*Figure 1—figure supplement 3*).

### HIV-1 DNA in untreated patients predicts disease progression

For this analysis, disease progression was defined according to the primary end-point of the SPARTAC trial, that is, a composite end-point of either a CD4 T cell count of 350 cells/µl or the commencement of long-term ART (for any clinical determined decision) (*SPARTAC Trial Investigators et al., 2013*). We carried out Kaplan–Meier survival analyses with patients randomised to receive no ART, and stratified

**Table 1.** Patient demographics

| | Total participants available for analysis* |
|---|---|
| Number | 154 |
| Patients with a Total HIV-1 DNA test | 154 (100%) |
| Patients with an Integrated HIV-1 test | 111 (72%) |
| Log10 baseline Total HIV-1 DNA copies/ml | 3.88 (3.42–4.24) |
| Log10 baseline Integrated HIV-1 DNA copies/ml | 3.6 (3.26–3.79) |
| Time since seroconversion (days) | 73.82 (49.2–95.8) |
| $Log_{10}$ Baseline Viral Load copies/ml | 4.62 (3.95–5.25) |
| Baseline CD4 Cell count (cells/µl) | 558 (428–680.9) |
| Country of recruitment | |
| Australia | 21 (13.6%) |
| Italy | 18 (12%) |
| Brazil | 13 (8.4%) |
| UK | 102 (66.2%) |
| Viral Subtype (%) | B (100%) |
| Sex | |
| Female | 4 (3%) |
| Male | 150 (97%) |

Data shown are values (% of non-missing values) for categorical data or medians and interquartile ranges in brackets for continuous variables.

*At the week 0 'baseline' timepoint'. A subset of these patients (**Supplementary files 1 and 2**) was used for analyses at later time-points.

according to median HIV-1 DNA level at time of recruitment (n = 51 for Total HIV-1 DNA, and n = 38 for Integrated [due to limited sample availability]) (patient demographics detailed in **Supplementary file 1**). There was a significant delay in clinical progression in those with lower Total and Integrated HIV-1 DNA at baseline (p = 0.0016 and 0.0022, respectively; log-rank test) (**Figure 2**). The median time from randomization to primary endpoint stratified by low and high Total HIV-1 DNA levels was 187.0 (IQR 127.0–222.0) and 77.9 (IQR 35.0–172.8) weeks, respectively, and for low and high Integrated levels was 187.7 (IQR 132.7–214.9) and 52.0 (IQR 32.4–161.3) weeks, respectively.

Univariable Cox analyses showed Total HIV-1 DNA (HR 4.16 per $log_{10}$ increase [CI 2.10–8.26]; p < 0.0001), Integrated HIV-1 DNA (HR 5.41 per $log_{10}$ increase (CI 1.65–18.04); p = 0.006) and plasma viral load (HR 1.74 per $log_{10}$ increase (CI 1.13–2.68) p = 0.011) predicted the trial primary endpoint (**Table 2**). Multivariable analyses were carried out with the baseline covariates, Total HIV-1 DNA, viral load, and CD4 T cell count. Here, Total HIV-1 DNA (HR = 3.57 (1.58–8.08); p = 0.002) and CD4 count (HR = 0.67 (0.53–0.84); p < 0.001), but not plasma viral load (HR = 1.25 (0.80–1.95); p = 0.33) predicted time to primary endpoint (**Table 2**). In a similar multivariable analysis, Integrated DNA did not associate significantly with the trial endpoint.

## HIV-1 DNA decline on ART

One third of the participants recruited to SPARTAC were randomised according to the trial protocol to receive 48 weeks of ART before undertaking a treatment interruption (**SPARTAC Trial Investigators et al., 2013**). This allowed us not only to study the impact of ART on HIV-1 DNA levels in this cohort (which has been reported in different cohorts [**Siliciano et al., 2003**; **Chun et al., 2007**]), but also to characterise what happens on stopping therapy after treatment initiated during PHI.

Prior to starting ART, Total and Integrated HIV-1 DNA levels were significantly different (p < 0.0001; Students t test) (**Figure 3**), most likely explained by the presence of unintegrated circular and linear DNA forms. As expected, HIV-1 DNA levels after 48 weeks of ART were significantly lower than those measured at baseline (p < 0.0001 for all comparisons; Students t test) by 0.63 log copies/million CD4 cells for Total, and 0.59 log copies/million CD4 cells for Integrated (**Figure 3**). After 48 weeks of ART, Total DNA levels remained significantly greater than Integrated levels in patients despite undetectable viraemia (0.027; paired t test) (**Figure 3**). This is consistent with other reports of residual unintegrated HIV-1 DNA up to a year after ART initiation (**Agosto et al., 2011**).

Having ascertained that in untreated individuals HIV-1 DNA was a predictor of progression, we now asked whether the lower HIV-1 DNA levels following ART would predict progression if therapy was stopped. This has potentially greater utility, as the majority of individuals on successful ART will have undetectable plasma viraemia using standard assays.

## HIV-1 DNA at the point of stopping ART predicts clinical progression

We measured DNA levels in participants who received a median of 48 (IQR 47.7–48.7) weeks of ART with successfully suppressed viraemia (VL < 50 copies/ml plasma), immediately prior to treatment

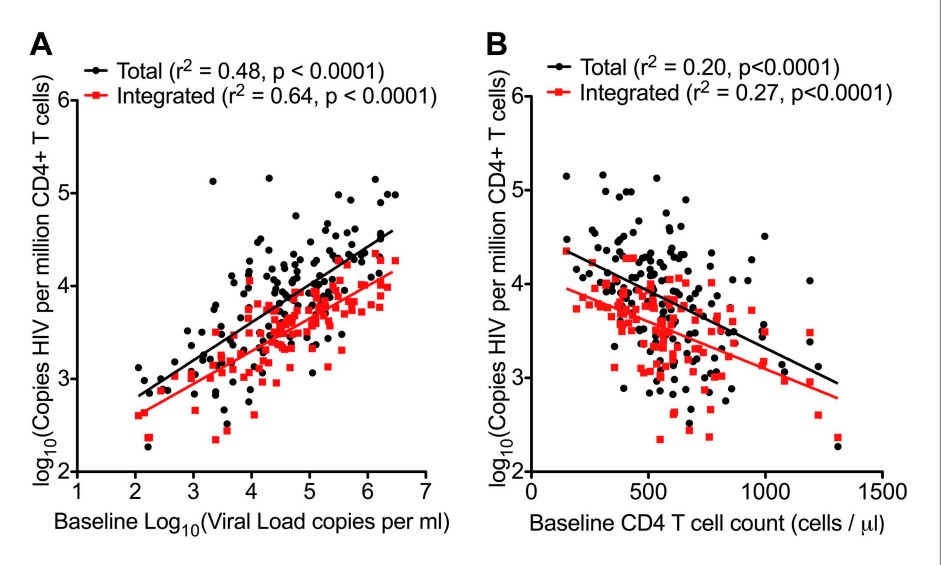

**Figure 1**. HIV-1 DNA correlates with baseline plasma viral load and CD4 T cell count. Pre-therapy 'baseline' Total HIV-1 DNA (black points and line) (n = 154) and Integrated HIV-1 DNA (n = 111) (red points and line) correlated with log10 plasma HIV-1 RNA (**A**) and CD4 cell count (**B**).

The following figure supplements are available for figure 1:

**Figure supplement 1**. Distribution of log10 total and integrated HIV-1-DNA levels in untreated patients at baseline.

**Figure supplement 2**. Pearson correlation for total and integrated HIV-1 DNA levels in untreated patients at baseline.

**Figure supplement 3**. Relationship between estimated time since seroconversion and HIV-1 DNA levels.

interruption. The demographics of the subset of individuals (n = 47) studied in this analysis are detailed in *Supplementary file 1*. Kaplan–Meier survival analyses were undertaken in which participants were again divided into two groups (low and high) based on median HIV-1 DNA levels at TI. Both low Total and Integrated HIV-1 DNA levels associated with a longer time to trial endpoint (p = 0.039 and 0.031, respectively; log-rank test) (*Figure 4*). The median time from TI to primary endpoint stratified by low and high Total HIV-1 DNA levels was 159.2 (IQR 111.9–200.6) and 117.8 (IQR 67.8–173.8) weeks, respectively, and by low and high Integrated levels was 166 (IQR 124.9–200.6) and 101.1 (IQR 65.5–156.8) weeks, respectively.

In univariable Cox regression analyses, Total and Integrated HIV-1 DNA both predicted clinical progression from TI, determined by time to reaching the trial primary endpoint (Total HR 3.52 [1.32–9.37]; p = 0.012; Integrated HR 3.01 (1.13–7.95); p = 0.027). Multivariable cox regression models were constructed with HIV-1 DNA and CD4 cell count at TI. Viral load was not included as it was undetectable at TI. Both Integrated (HR 2.81 CI (1.05–7.55) p = 0.04) and Total (HR 3.42 CI (1.29–9.05) p = 0.013) HIV-1 DNA retained significance, and in both cases CD4 T cell count at TI was not a significant predictor (HR 1.04 CI (0.83–1.11) p = 0.58 and HR 0.94 CI 0.825–1.08 p = 0.4). At TI, HIV-1 DNA was the only predictor of the primary end point.

## HIV-1 DNA increases on stopping ART

One of the concerns around the viral rebound following a TI is the risk of 're-seeding' the reservoir in individuals who might have extremely low HIV-1 DNA levels, and who might be candidates for 'post-treatment control' of viraemia (*Hocqueloux et al., 2010*). We therefore measured HIV-1 DNA in those participants who had received 48 weeks of ART at the point of TI and then again 4, 12 and 60 weeks post TI, where samples were available. Total and Integrated HIV-1 DNA levels were not significantly

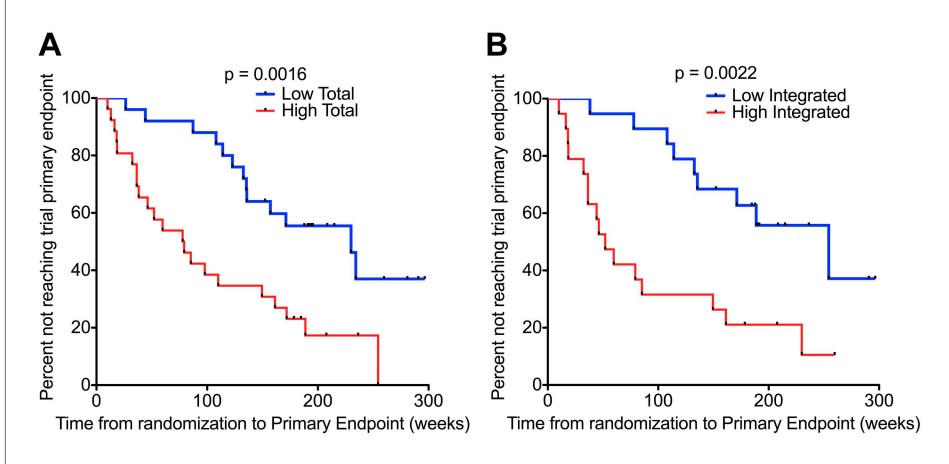

**Figure 2**. HIV-1 DNA predicts clinical progression in absence of ART. Kaplan–Meier survival analyses for (**A**) Total (n = 51) and (**B**) Integrated (n = 38) HIV-1 DNA and clinical progression, based on time from randomization to the SPARTAC trial primary endpoint of a CD4 T cell count of 350 cells/µl or starting long-term ART. HIV-1 DNA data was divided into two 'high' and 'low' at the median level, which was 4.02 and 3.61 copies HIV-1 DNA per million CD4 T cells for Total and Integrated, respectively. Significance was determined by log rank test.

greater than at the time of ART cessation for up to 12 weeks post TI, although had significantly increased 60 weeks after TI (p < 0.0001 for Total and Integrated DNA; Students $t$ test), returning approximately to the Week 0 pre-therapy levels (*Figure 3*). The increase in Total and Integrated HIV-1 DNA 4 weeks after TI was not significant (p = 0.30), in contrast to the rebound in plasma viraemia (p < 0.001), which may be re-assuring for those implementing a TI strategy in which ART would be re-introduced when plasma VL became detectable.

Of note, in an analysis of those individuals who subsequently restarted ART after the TI–and for whom we had samples (n = 15)–there was no significant difference between the HIV-1 reservoir size pre-TI and at least 6 months after re-starting ART (p = 0.58; paired students $t$ test; *Figure 3—figure supplement 1*), suggesting that any increase in HIV-1 DNA on stopping ART may be reversible if therapy is re-commenced. However, larger studies will be needed to confirm these data.

## HIV-1 DNA at ART cessation predicts time to plasma viral load rebound

Although almost all participants in SPARTAC experienced VL rebound on stopping ART, we have previously shown that of those who received >12 weeks of therapy, 14% still had undetectable viraemia 12 months later (*Stohr et al., 2013*). We therefore wished to establish—albeit in this different, although overlapping, sub-group of SPARTAC participants—whether HIV-1 DNA predicted the return of plasma viraemia post-TI. As our previous findings included participants in centres using both 50 and

**Table 2.** Cox regression models for variables associated with clinical progression in untreated individuals followed up from PHI

| Univariable unadjusted | | | Multivariable adjusted | |
| --- | --- | --- | --- | --- |
| Covariate | HR (95% CI) | p Value | HR (95% CI) | p Value |
| Total DNA (log$_{10}$ DNA copies) | 4.16 (2.1–8.26) | <0.001 | 3.57 (1.58–8.08) | 0.002 |
| Viral load (log$_{10}$ RNA copies) | 1.74 (1.14–2.67) | 0.011 | 1.25 (0.80–1.95) | 0.33 |
| CD4+ T cell count/100 cells | 0.66 (0.53–0.82) | <0.001 | 0.67 (0.53–0.84) | <0.001 |

Univariable and multivariable cox regression models were used to determine predictors of clinical progression in untreated individuals followed up from Primary HIV-1 Infection. Progression was determined according to reaching the SPARTAC trial primary endpoint (*Chun et al., 2010*). Co-variables analysed were baseline (i.e. first pre-therapy trial sample) Total HIV-1 DNA, baseline plasma viral load and baseline CD4+ T cell count.

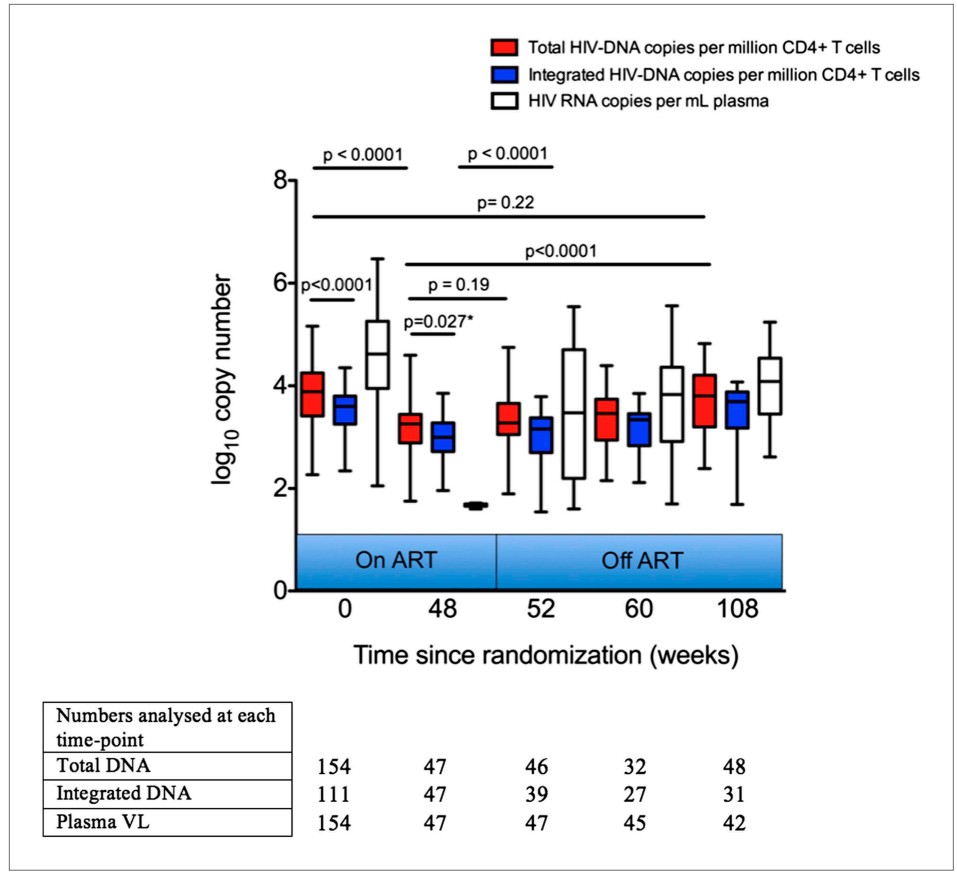

**Figure 3**. Analysis of impact on HIV-1 DNA of antiretroviral therapy. Total and Integrated HIV-1 DNA levels and plasma viral load (HIV-1 RNA) were measured at Week 0 'baseline' (in participants from all three trial arms prior to any therapy) and also in those receiving 48 weeks of ART (weeks 48, 52, 60 and 108 after baseline). DNA levels (log10 copies/million CD4 T cells) and viral load (log10 copies/ml plasma) were measured at all time-points, but not all participants were sampled at all time-points dependent on sample availability. Significance was determined by unpaired Students $t$ Tests or paired $t$ test (marker with *) when samples at the two time-points being compared were matched.

The following figure supplement is available for figure 3:

**Figure supplement 1**. Impact of stopping and re-starting ART on HIV-1 DNA.

400 copies/ml as the lower limit of detection for plasma viral load assays, we studied both cut-offs for the HIV-1 DNA analyses.

No patients were censored before viral rebound was detected and all were aviraemic (<50 copies/ml plasma) at the point of stopping ART. Levels of Total (but not Integrated) HIV-1 DNA at TI predicted time to viral rebound to 400 copies/ml by univariable Cox regression analysis (HR 2.43 (1.23–4.79) p = 0.010). CD4 T cell count at TI was not predictive (HR 0.92 (0.78–1.08) p = 0.32) (*Supplementary file 3*). In a multivariable Cox regression model including Total HIV-1 DNA and CD4 count, both sampled at the point of TI, only Total HIV-1 DNA significantly predicted time to viral rebound to 400 copies (HR 2.68 [1.31–5.48] p = 0.0069) (*Supplementary file 3*). When using values from pre-therapy baseline rather than at the time of TI in the model, neither plasma viral load nor CD4 T cell count predicted time to viral rebound (>400 copies/ml) from TI (HR 1.38 [0.96–1.99] p = 0.080) and (HR 1.03 [0.92-1.12] p = 0.60), respectively. Kaplan–Meier survival analyses showed similar results, with a low Total HIV-1 DNA (based on stratification around the median level) associated with a slower time to a viral rebound of 400 copies/ml (p = 0.0038; log-rank test) but not to 50 copies per ml (p = 0.18) (*Figure 5*).

It was unclear why Total HIV-1 DNA should predict rebound to 400 copies but not to 50. In an attempt to explain this we studied those individuals with data available at both cut-offs. In this small

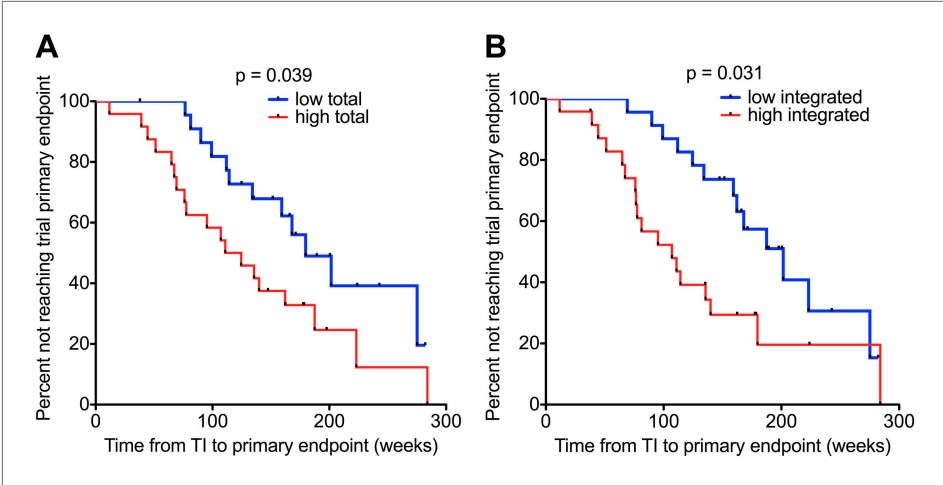

**Figure 4**. HIV-1 DNA on ART predicts clinical progression following treatment interruption. Kaplan–Meier survival analyses for (**A**) Total (n = 47) and (**B**) Integrated (n = 47) HIV-1 DNA and clinical progression, based on time to the SPARTAC trial primary endpoint of a CD4 T cell count of 350 cells/μl or starting back on long-term ART. HIV-1 DNA data was divided into 'high' and 'low' at the median. Significance was determined by log rank test. Participants had received a median of 48 weeks of ART and then undertook a treatment interruption. DNA levels were measured at week 48, at the point of stopping ART. Time from TI to primary endpoint is plotted on the x-axis.

post-hoc analysis (n = 45), we found that rebound varied according to the HIV-1 DNA level at the time of TI. Patients with high Total HIV-1 DNA levels were more likely to have a first detectable VL greater than 400 copies/ml, whereas those with lower HIV-1 DNA levels were more likely initially to rebound below 400 but above 50 copies/ml (p = 0.0074; Fisher's exact test) (**Supplementary file 4**). In summary, we find evidence that HIV-1 DNA is a significant predictor of the duration of viral remission and magnitude of the initial rebound following TI. This, if confirmed in larger studies, would have implications for those designing protocols for ART-reintroduction following viral rebound in TI studies.

## Discussion

Since first described nearly two decades ago a persistent reservoir of HIV-1-infected cells remains the main reason that HIV-1 infection cannot be cured (*Chun et al., 1997*; *Finzi et al., 1997*; *Wong et al., 1997*). The simplest measure of the reservoir is a qPCR assay that detects all intracellular HIV-1 DNA regardless of whether it is integrated into host chromosomes or is in unintegrated linear or circular forms. A modification of this assay incorporates an initial step to prime host *Alu* repeats in order to quantify only viral DNA that has been integrated into host DNA. These assays are open to criticism as the vast majority of intracellular HIV-1 DNA is thought to be replication incompetent, and qPCR is not able to discriminate between replication competent and incompetent viral DNA genomes. This has led to the development of alternative approaches such as viral outgrowth assays (which are considered the gold standard, but are expensive and time-consuming, even with recent improvements to their protocols [*Laird et al., 2013*]) and assays to measure intracellular HIV-1 RNA, which may more accurately reflect an infected cell's ability to produce new virions, especially under conditions where viral transcription is stimulated (*Bullen et al., 2014*).

Despite the debate over the biological relevance of measuring HIV-1 DNA—and bearing in mind that none of these assays have been standardised for clinical use–a number of reports have attributed clinical meaning to HIV-1 DNA assays. Over a decade ago Tierney and colleagues suggested that proviral DNA in PBMCs from 111 participants receiving limited nucleoside analogue therapy was an independent predictor of clinical progression, although it is unclear how suppressive the ART regimes were in this study (*Tierney et al., 2003*). Havlir et al studied 100 individuals with chronic HIV-1 infection and viral suppression on ART and showed that HIV-1 DNA independently predicted residual viraemia on ART (*Havlir et al., 2005*). However there has not been a comprehensive analysis of both HIV-1 Total and Integrated HIV-1 DNA in individuals randomised to treatment or no treatment soon after seroconversion.

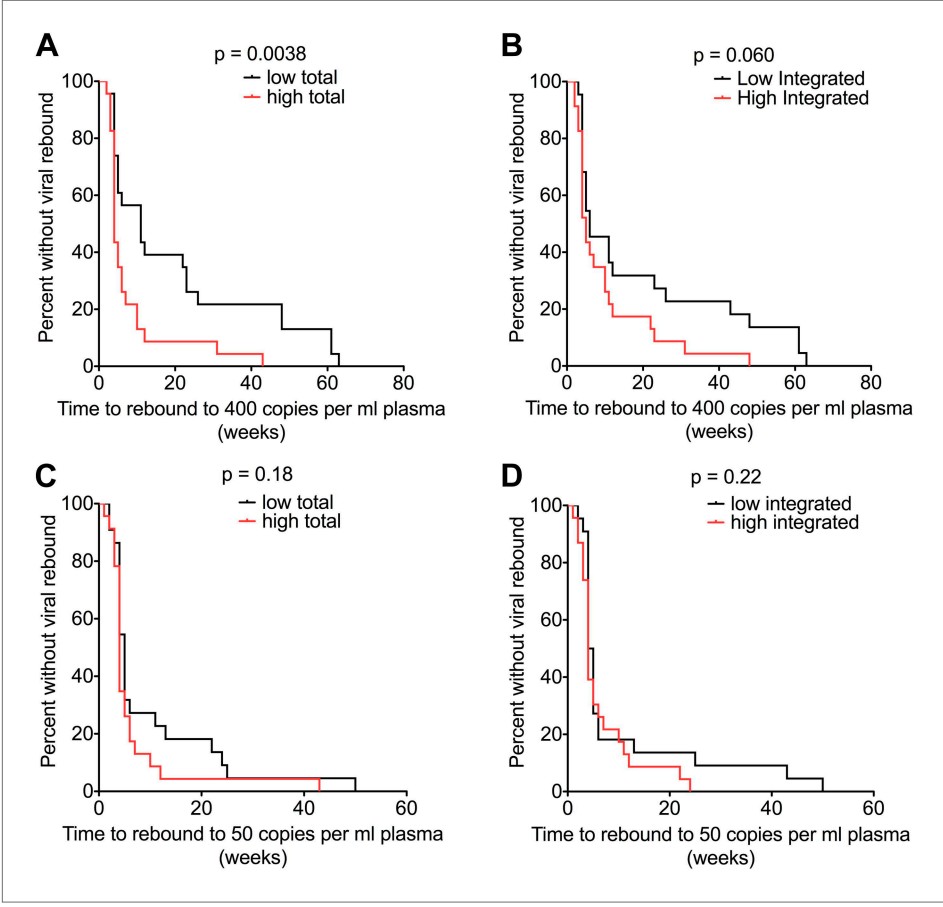

**Figure 5**. HIV-1 DNA at ART interruption predicts time to viral rebound. Survival analyses of time to viral rebound (weeks) in participants undertaking TI after 48 weeks of ART. HIV-1 DNA levels are presented divided at the median level into high (red) and low (black). Rebound to 400 HIV-1 RNA copies (n = 46) is presented for Total (**A**) and Integrated (**B**) HIV-1 DNA. Rebound to 50 HIV-1 RNA copies (n = 45) is presented for Total (**C**) and Integrated (**D**) HIV-1 DNA. Significance was determined by log rank test.

We applied both Total and Integrated DNA measures to a unique cohort of individuals with evidence of PHI randomised to immediate interrupted ART or no therapy with longitudinal follow-up for a median of 4.5 years. As participants were randomised to different short course ART therapies prior to TI, we were able to determine how well HIV-1 DNA correlated with accepted surrogate markers of progression such as VL and CD4 count, and also whether HIV-1 DNA was an independent predictor of disease progression within the SPARTAC trial in both treated and untreated participants.

Our first finding that HIV-1 DNA associated closely with both plasma VL and CD4 cell counts (*Figure 1*) was not surprising as this is reported elsewhere (*Chun et al., 2010*; *Parisi et al., 2012*). Our findings that both baseline and pre-TI HIV-1 DNA strongly predicted the trial primary endpoint (*Figure 4* and *Table 2*) are supported by data from other smaller, discrete observational studies, in which low HIV-1 DNA levels associated with a longer time to clinical progression (*Goujard et al., 2006*; *Minga et al., 2008*; *Piketty et al., 2010*), a lower viral set point and reduced chance of virological failure on ART re-initiation (*Yerly et al., 2004*) at PHI. We are aware of one other report associating HIV-1 DNA with time to viral rebound on stopping ART (*Yerly et al., 2004*). In this study Yerly and colleagues studied chronically-infected individuals with sequential treatment interruptions and reported that DNA was a predictor of the peak of viraemia following therapy cessation and failure to reach undetectable viraemia on re-starting ART–they do not report on the actual duration of viral suppression after TI. In a smaller study at PHI, Lafeuillade et al also associated HIV-1 DNA with time to rebound, however this study is complicated by other interventions such as IL-2 and hydroxyurea in addition to ART (*Lafeuillade et al., 2003*).

We measured plasma viral load in the pre-therapy 'baseline' sample closest to the estimated time of infection. One possible criticism—and explanation for why plasma VL was less predictive in this study—is that other studies have associated progression with the 'set-point' viral load, the value at which the VL stabilizes following the dynamic PHI stage. However, in our untreated participants we found that the 'baseline' and 'set-point' VL values were highly correlated, although the former was higher, as would be expected (data not shown). From a clinical perspective, it is worth noting that if individuals with PHI are commenced on ART immediately, then their 'set-point' VL will not be known, potentially placing greater impact on the less dynamic HIV-1 DNA measure.

After TI, we observed a period of at least 12 weeks where no significant increases in the HIV-1 reservoir level were detected by both assays (*Figure 3*). However, we found little evidence of longer term post-treatment control (*Persaud et al., 2013*; *Saez-Cirion et al., 2013*), as levels of HIV-1 DNA 1-year after therapy interruption were not significantly different to that seen at pre-therapy baseline. Nevertheless, the potential for there to be a short window period during which plasma viraemia has rebounded but HIV-1 DNA levels have not risen significantly is encouraging, if future closely-monitored TI studies are to be undertaken. Concerns around 're-seeding' the reservoir are very real, and it is important that any possible harm associated with a TI is limited. It is therefore also re-assuring that in our admittedly small sub-study, re-initiation of ART subsequently restored HIV-1 DNA to pre-TI levels.

Finally, a low 'Total' HIV-1 reservoir at TI resulted in a longer time to a viral rebound to 400 copies/ml (*Figure 5*). In univariable and multivariable Cox regression models Total HIV-1 DNA at TI predicted time to rebound to 400 copies/ml, whereas CD4 T cell count did not (*Supplementary file 3*). Of interest, the baseline VL and CD4 prior to therapy were also not predictive of time to rebound. In contrast to other studies exploring TI, we have a larger and randomly allocated patient group who have received similar durations of ART at PHI and hence can be directly compared. Although viral rebound was observed in all individuals after TI ultimately, this is the first report of a randomised cohort that has shown that time to viral rebound and primary study end point could be predicted by HIV-1 DNA measurement at TI.

Our findings of an association with HIV-1 DNA and time to viral rebound raise a number of other questions. Why was Total DNA predictive of rebound but not Integrated? Why was Total DNA predictive for rebound to 400 copies/ml but not 50 copies/ml? Much larger studies will be needed to answer most of these questions, however our sub-analysis of rebounding patients suggested that a high Total DNA at TI was more indicative of a higher VL rebound (i.e. >400 copies), whereas a low DNA level was not associated with a lower rebound. These data might indicate that a Total DNA level at TI is better at predicting the patients who will be quick to rebound rather than those who will maintain suppression. A question for larger studies to answer will be to define what the viral load cut-off should be for considering rebound, rather than just assuming the assay with the lowest limit of detection is best. Data from at least one other study (*Riabaudo et al., 2009*) indicate that a level greater than 50 copies/ml may be more relevant. The difference between the Integrated and Total DNA is also interesting. Integrated DNA should be the most biologically relevant marker, based on the assumption that unintegrated HIV-1 DNA forms are thought not to contribute to rebound viraemia. However, the assay for Total HIV-1 DNA is much simpler and with tighter coefficients of variation, possibly due to the lack of a pre-amplification PCR stage. Another important factor impacting our data is that the median estimated time from seroconversion was 73.8 days, and so most of our patients would be starting therapy at Fiebig stage IV or later. It is possible that earlier identification of PHI and initiation of ART would have a greater impact on the reservoir and post-treatment control, and it is important that large studies are undertaken to determine this.

In light of observational cohorts such as VISCONTI (*Saez-Cirion et al., 2013*) where treatment cessation revealed individuals who remain aviraemic post TI, there is increasing interest in undertaking closely monitored treatment interruption studies in which ART would be re-started based on a detectable plasma VL. These do not, however, have an encouraging history with previous studies set in the context of therapeutic vaccination or CD4 T cell restoration, resulting in rapid viraemic rebound and even harm (*Strategies for Management of Antiretroviral Therapy (SMART) Study Group et al., 2006*; *Angel et al., 2011*; *Garcia et al., 2013*). Additionally, the recent report of viral rebound in the case of the Mississippi baby, means that greater understanding of mechanisms behind post-treatment control is needed. The potential, therefore, to develop an algorithm to combine various biomarkers to help predict individuals suitable for such studies is appealing. These data are evidence that such an algorithm may be possible, and that a marker as simple as HIV-1 DNA could be an important component.

## Materials and methods

### Participants and trial design

The design of the SPARTAC trial is reported elsewhere (*SPARTAC Trial Investigators et al., 2013*). In brief, SPARTAC was an international open Randomised Controlled Trial enrolling adults with PHI within 6 months of a last negative, equivocal or incident HIV-1 test. All participants gave written informed consent. Research ethics committees in each country approved the trial. Time of seroconversion was estimated as the midpoint of last negative/equivocal and first positive tests, or date of incident test. Participants were randomised to receive ART for 48 weeks (ART-48), 12 weeks (ART-12) or no therapy (standard of care, SOC). The primary endpoint was a composite of two events: if participants either reached a CD4 count of <350 cells/mm$^3$ (>3 months after randomization and confirmed within 4 weeks) or initiated long-term ART. This provided an immunological surrogate of clinical progression, but also allowed inclusion of those participants who commenced ART at CD4 cell counts greater than 350 cells/mm$^3$. Time to virological failure of participants randomized to ART-48 (two analyses using both 50 and 400 HIV-1 RNA copies/ml as the cut-off [two consecutive readings]) was a secondary endpoint.

Participants for this sub-study of SPARTAC were those infected with subtype B HIV-1 and for whom adequate samples were available. For those in the analysis of progression and viral rebound at TI, we only selected participants who had viral load suppression (<50 copies/ml; Chiron bDNA) at point of stopping ART (*Table 1* and *Supplementary file 1*). CD4 T cells isolated from peripheral blood mononuclear cells (PBMC) were sampled for HIV-1 DNA in all participants at baseline, regardless of trial arm. Participants randomised to the ART-48 arm were sampled at week 48 at the point of stopping ART and at a further 4, 12 and 60 weeks post ART interruption (52, 60 and 108 weeks post-ART initiation). Participants who were viraemic using the Chiron bDNA, (Bayer, Leverkusen, Germany) (LLD 50 copies/ml) at the point of TI were excluded.

### Measurement of HIV-1 DNA

CD4 T cells were enriched from frozen PBMC samples by negative selection (Dynabeads, Invitrogen, Carlsbad, CA) to a purity of >97%. CD4 T cell DNA was extracted (Qiagen, Venlo, Netherlands) and used as input DNA for PCR. Cell copy number and total HIV-1 DNA levels were quantified both in triplicate using previously published assays (*Duncan et al., 2013*; *Jones et al., 2014*).

Integrated HIV-1 was measured using an assay based on that previously published (*Liszewski et al., 2009*) but with some minor modifications. 40 repeated integration measurements per patient sample were performed along with five PCR reactions to which no *Alu* primer was added, to serve as a background control for determination of sample positivity. The first round master mix contained 1.5 U platinum taq per 50 μl reaction. The second round qPCR reaction was the same as the Total reaction described above, but with 10 μl of first round product being the input DNA.

To quantify patient samples, one standard curve was generated by plotting the average cycle threshold (Ct) values for all integration signals at each Integration Standard (IS) dilution (70–0.2 copies of IS standard per well, diluted in 2 μg/ml PBMC DNA), so long as at least one integration signal was significantly different (two standard deviations) to the average *gag*-only background signal. The IS was a kind gift from Una O'Doherty. Ln(Copy number) was plotted vs Ln(average Ct) and each point on the standard curve was repeated in duplicate. The standard curve fitted extremely well to a line of best fit ($r^2 = 0.987$), which was then used to calculate copy numbers in patient samples. Each patient sample replicate was quantified individually using the standard curve to generate error. Plate to plate variation was assessed using quadruplicate replicates of 8e5 cells, which have one copy of HIV-1 per cell, diluted to 100 copies per well as a first round PCR DNA input. The average coefficient of variance was 8.31%.

### Statistical analysis

HIV-1 DNA (Total and Integrated) distributed normally following log$_{10}$ transformation. The association between Total and Integrated HIV-1 DNA levels was tested using Pearson correlations. Linear regression was used to examine the association between continuous clinical baseline covariates and HIV-1 DNA. Tests between grouped variables and DNA levels were tested with Mann–Whitney, Kruskal–Wallis and *t* tests where appropriate.

For the association between baseline DNA and the SPARTAC primary endpoint, Kaplan–Meier plots and univariable Cox models were constructed, and subsequently adjusted for baseline covariates. Where participants received ART, the time to primary endpoint was calculated from the time of TI.

Association with time to rebound was also assessed using Kaplan–Meier plots and Cox models. All statistics were calculated using R version 3.1.0. Plots were drawn using Prism version 5.0.

## Acknowledgements

We thank the participants of SPARTAC and the SPARTAC Trial Investigators: Trial Steering Committee: Independent Members- A Breckenridge (Chair), P Clayden, C Conlon, F Conradie, J Kaldor*, F Maggiolo, F Ssali, Country Principal Investigators- DA Cooper, P Kaleebu, G Ramjee, M Schechter, G Tambussi, JM Miro, J Weber. Trial Physician: S Fidler. Trial Statistician: A Babiker. Data and Safety Monitoring Committee (DSMC): T Peto (Chair), A McLaren (in memoriam), V Beral, G Chene, J Hakim. Co-ordinating Trial Centre: Medical Research Council Clinical Trials Unit, London (A Babiker, K Porter, M Thomason, F Ewings, M Gabriel, D Johnson, K Thompson, A Cursley*, K Donegan*, E Fossey*, P Kelleher*, K Lee*, B Murphy*, D Nock*). Central Immunology Laboratories and Repositories: The Peter Medawar Building for Pathogen Research, University of Oxford, UK (R Phillips, J Frater, L Ohm Laursen*, N Robinson, P Goulder, H Brown). Central Virology Laboratories and Repositories: Jefferiss Trust Laboratories, Imperial College, London, UK (M McClure, D Bonsall*, O Erlwein*, A Helander*, S Kaye, M Robinson, L Cook*, G Adcock*, P Ahmed*). Clinical Endpoint Review Committee: N Paton, S Fidler. Investigators and Staff at Participating Sites: Australia: St Vincents Hospital, Sydney (A Kelleher), Northside Clinic, Melbourne (R Moore), East Sydney Doctors, Sydney (R McFarlane), Prahran Market Clinic, Melbourne (N Roth), Taylor Square Private Clinic, Sydney (R Finlayson), The Centre Clinic, Melbourne (B Kiem Tee), Sexual Health Centre, Melbourne (T Read), AIDS Medical Unit, Brisbane (M Kelly), Burwood Rd Practice, Sydney (N Doong), Holdsworth House Medical Practice, Sydney (M Bloch), Aids Research Initiative, Sydney (C Workman). Coordinating Centre in Australia: Kirby Institute University of New South Wales, Sydney (P Grey, DA Cooper, A Kelleher, M Law). Brazil: Projeto Praca Onze, Hospital Escola Sao Francisco de Assis, Universidade federal do Rio de Janeiro, Rio de Janeiro (M Schechter, P Gama, M Mercon*, M Barbosa de Souza, C Beppu Yoshida, JR Grangeiro da Silva, A Sampaio Amaral, D Fernandes de Aguiar, M de Fatima Melo, R Quaresma Garrido). Italy: Ospedale San Raffaele, Milan (G Tambussi, S Nozza, M Pogliaghi, S Chiappetta, L Della Torre, E Gasparotto), Ospedale Lazzaro Spallanzani, Roma (G DOffizi, C Vlassi, A Corpolongo). South Africa: Cape Town: Desmond Tutu HIV-1 Centre, Institute of Infectious Diseases, Cape Town (R Wood, J Pitt, C Orrell, F Cilliers, R Croxford, K Middelkoop, LG Bekker, C Heiberg, J Aploon, N Killa, E Fielder, T Buhler). Johannesburg: The Wits Reproductive Health and HIV-1 Institute, University of Witswatersrand, Hillbrow Health Precinct, Johannesburg (H Rees, F Venter, T Palanee), Contract Laboratory Services, Johannesburg Hospital, Johannesburg (W Stevens, C Ingram, M Majam, M Papathanasopoulos). Kwazulu-Natal: HIV-1 Prevention Unit, Medical Research Council, Durban (G Ramjee, S Gappoo, J Moodley, A Premrajh, L Zako). Uganda: Medical Research Council/Uganda Virus Research Institute, Entebbe (H Grosskurth, A Kamali, P Kaleebu, U Bahemuka, J Mugisha*, HF Njaj*). Spain: Hospital Clinic-IDIBAPS, University of Barcelona, Barcelona (JM Miro, M Lopez-Dieguez*, C Manzardo, JA Arnaiz, T Pumarola, M Plana, M Tuset, MC Ligero, MT Garca, T Gallart, JM Gatell). UK and Ireland: Royal Sussex County Hospital, Brighton (M Fisher, K Hobbs, N Perry, D Pao, D Maitland, L Heald), St Jamess Hospital, Dublin (F Mulcahy, G Courtney, S ODea, D Reidy), Regional Infectious Diseases Unit, Western General Hospital and Genitourinary Dept, Royal Infirmary of Edinburgh, Edinburgh (C Leen, G Scott, L Ellis, S Morris, P Simmonds), Chelsea and Westminster Hospital, London (B Gazzard, D Hawkins, C Higgs), Homerton Hospital, London (J Anderson, S Mguni), Mortimer Market Centre, London (I Williams, N De Esteban, P Pellegrino, A Arenas-Pinto, D Cornforth*, J Turner*), North Middlesex Hospital (J Ainsworth, A Waters), Royal Free Hospital, London (M Johnson, S Kinloch, A Carroll, P Byrne, Z Cuthbertson), Barts and the London NHS Trust, London (C Orkin, J Hand, C De Souza), St Marys Hospital, London (J Weber, S Fidler, E Hamlyn, E Thomson*, J Fox*, K Legg, S Mullaney*, A Winston, S Wilson, P Ambrose), Birmingham Heartlands Hospital, Birmingham (S Taylor, G Gilleran). Imperial College Trial Secretariat: S Keeling, A Becker. Imperial College DSMC Secretariat: C Boocock.

  * Left the study team before the trial ended.

  We thank Dr José M Miro and Dr Juan Ambrosioni for helpful comments on the manuscript. We thank Una O'Doherty for the Alu-PCR Integration Standard and guidance on assay development. The SPARTAC trial was funded by a grant from the Wellcome Trust (069598/Z/02/Z). JF is supported by the Medical Research Council, the National Institute for Health Research Oxford Biomedical Research Centre, and the Oxford Martin School. This project has been funded in part with federal funds from the Frederick National Laboratory for Cancer Research, under Contract No. HHSN261200800001E. The content of this publication does not necessarily reflect the views or policies of the Department of

Health and Human Services, nor does mention of trade names, commercial products, or organizations imply endorsement by the U.S. Government. This Research was supported in part by the Intramural Research Program of the NIH, Frederick National Lab, Center for Cancer Research.

## Additional information

### Funding

| Funder | Grant reference number | Author |
|---|---|---|
| Oxford Martin School, University of Oxford | Institute of Emerging Infections | Rodney E Phillips, John Frater |
| National Institute of Health Research, Oxford BRC | Oxford BRC Immunology Theme | John Frater |
| Nuffield Department of Medicine | NDM Studentship | James P Williams |
| Wellcome Trust | SPARTAC Trial Grant:(069598/Z/02/Z) | David Cooper, Abdel Babiker, Jonathan Weber, Rodney E Phillips |

The funders had no role in study design, data collection and interpretation, or the decision to submit the work for publication.

### Author contributions

JPW, JH, JF, Conception and design, Acquisition of data, Analysis and interpretation of data, Drafting or revising the article, Contributed unpublished essential data or reagents; WS, Analysis and interpretation of data, Drafting or revising the article; NR, HB, Acquisition of data, Analysis and interpretation of data; MF, SK, GT, Conception and design, Acquisition of data, Drafting or revising the article; DC, SF, Conception and design, Acquisition of data, Analysis and interpretation of data, Drafting or revising the article; MS, AB, JW, KKK, ADK, REP, Conception and design, Analysis and interpretation of data, Drafting or revising the article; MC, Analysis and interpretation of data, Drafting or revising the article, Contributed unpublished essential data or reagents

### Ethics

Clinical trial Registry: NCT. Registration ID: NCT00912041.
Clinical trial Registry: EudraCT. Registration ID: EudraCT2004-000446-20.
Human subjects: The SPARTAC trial was approved by the following authorities: Medicines and Healthcare products Regulatory Agency (UK), Ministry of Health (Brazil), Irish Medicines Board (Ireland), Medicines Control Council (South Africa), and The Uganda National Council for Science and Technology (Uganda). It was also approved by the following ethics committees in the participating countries: Central London Research Ethics Committee (UK), Hospital Universitario Clementino Fraga Filho Ethics in Research Committee (Brazil), Clinical Research and Ethics Committee of Hospital Clinic in the province of Barcelona, Spain, The Adelaide and Meath Hospital Research Ethics Committee (Ireland), University of Witwatersrand Human Research Ethics Committee, University of Kwazulu-Natal Research Ethics Committee and University of Cape Town Research Ethics Committee (South Africa), Uganda Virus Research Institute Science and ethics committee (Uganda), The Prince Charles Hospital Human Research Ethics Committee and St Vincent's Hospital Human Research Ethics Committee (Australia), and the National Institute for Infectious Diseases Lazzaro Spallanzani, Institute Hospital and the Medical Research Ethics Committee, and the ethical committee Of the Central Foundation of San Raffaele, MonteTabor (Italy). All participants signed a written informed consent.

## Additional files

### Supplementary files

• Supplementary file 1. Additional demographics of randomized participants included in untreated and 48 week short-course ART analyses Demographics of participants available for analyses of those

randomised to receive either no therapy from PHI (first column) and those randomised to receive 48 of weeks of ART from PHI (second column). Data as indicated were: † determined at pre-therapy baseline (trial week 0), * determined at week 48, prior to TI or + median (interquartile range). SOC: Standard of Care trial arm.

• Supplementary file 2. Sample numbers available at each time-point by trial randomization. Numbers of samples available at each time-point are presented. Participants from all three trial arms were included at week 0 as they were all treatment naïve at this point. Not all patients at any one time-point are always represented at other time-points due to variation in sample availability. Trial arms: SOC: Standard of Care (untreated); ART-48: 48 weeks of ART after randomization; ART-12: 12 weeks of ART after randomization.

• Supplementary file 3. Cox regression models for variables associated with time to rebound of 400 copies/ml and sampled at wk48. Table to show results of Cox regression analysis for time to virological rebound of 400 copies/ml of plasma with Total DNA and CD4 T cell count as covariables. Univariable and multivariable data are presented with Hazard Ratios (HR) with 95% Confidence Intervals (CI) and associated P values.

• Supplementary file 4. 2 × 2 table comparing the number of patients with different times to 50 and 400 copy/ml rebound and their total and integrated pre-TI HIV-1 DNA levels. Table to compare the association between HIV-1 DNA levels at TI (both Total and Integrated) and time to a plasma viral load of either between 50–400 copies/ml or greater than 400 copies/ml. HIV-1 DNA levels were split into 'high' and 'low' by the median value. The proportions are significantly different by Fisher's exact test for Total (p = 0.0074) but not Integrated HIV-1-DNA levels (p = 0.091).

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
