## [Decision Letter]

Thank you for sending your work entitled ‘HIV-1 DNA predicts disease progression and post-treatment virological control’ for consideration at *eLife*. Your article has been favorably evaluated by Prabhat Jha (Senior editor), a Reviewing editor, and 3 reviewers.

The Reviewing editor and the other reviewers discussed their comments before we reached this decision, and the Reviewing editor has assembled the following comments to help you prepare a revised submission.

There is consensus that this manuscript is impressive and worth publishing. There is also agreement that while overall this manuscript is well written it is too long and too detailed and is distracting from the key and novel scientific advance. Careful editing of established and already published aspects and referencing with existing literature is needed. The Discussion also can be shortened and authors should stay focused on the novel and important findings. The novel finding is the predictive value of HIV DNA and time to viral rebound and keeping the focus on this will strengthen the manuscript. The finding is important as it can be used as a predictive marker in advancing the HIV cure research agenda. This finding and its significance is lost in the detailed presentation and discussion of novel and not so novel data. Reviewer comments build around this novel finding for a more nuanced picture to emerge in terms of request for more details on measurement of DNA, for example an explanation on the somewhat paradoxical situation as to why free DNA is more important than integrated DNA and to the analysis of the DNA as a continuous variable rather than as a dichotomy.

A separate issue relates to new data on the Mississippi baby that was announced subsequent to the submission of this manuscript, namely the baby has detectable virus and this manuscript needs to be updated accordingly.

Minor comments:

The use of the term ‘reservoir’ when measuring HIV DNA will not be well-received by many in the field. This term should be used sparingly, or the authors should clarify what they mean by this term.

In the Introduction, the authors suggest that the VISCONTI cohort argue against using HIV DNA as a predictor of what happens in absence of therapy. This may be true when compared to possible cure cases, but not when compared to the general HIV-infected population. The VISCONTI cohort is notable for having had a very low amount of HIV DNA, comparable to ‘elite’ controllers. The French team has argued that low DNA contributed to the outcome in this cohort. This text might be modified accordingly.

The inverse correlation between CD4+ T cell count and frequency HIV DNA in CD4+ T cells suggests that the total burden of HIV (per unit of blood) may be relatively stable. This brings up a constant dilemma with these cell-based assays: should the frequency of infection or total amount of infection be used in the analyses. Assessing whether the amount of HIV DNA in blood rather than frequency of HIV DNA in CD4+ T cells might provide unique insights, and might be considered for this paper, or future analyses.

The association of HLA and immune responses on HIV replication has been the focus of decades worth of research. Given the strong association between HIV RNA and HIV DNA in untreated state, it is not surprising that immune response might be negatively correlated with HIV DNA during pre-therapy state. The functional assay used (ELISpot interferon-gamma) is less robust and informative of most assays now being applied to these types of studies. The data as presented are fine, but it makes it more challenging for a reader to find the more innovative data.

Presumably the association between immune response (HLA, ELIspot) and HIV DNA was similar to that observed between immune response and HIV RNA levels. This might be tested. Again, it is unclear if there is really much new in this analysis. The fact that HIV DNA predicted disease progression independent of HIV RNA is novel, and supports doing a similar analysis with the immunologic and genetic data.

The paper is generally easy to follow. The last sentence in the Results is not clear, however. The authors might wish to restate these findings.

---

## [Author Response]

*There is consensus that this manuscript is impressive and worth publishing. There is also agreement that while overall this manuscript is well written it is too long and too detailed and is distracting from the key and novel scientific advance. Careful editing of established and already published aspects and referencing with existing literature is needed*.

We have now completely removed the sections on CD4 and CD8 ELISpots as well as the sections on HLA Class I associations and related hazards of disease progression. This means the paper is much more focused on the data associated with HIV-1 DNA quantification. As a result the paper has been shortened from 5736 words to 4699 words (around 20% shorter) including the removal of Figures 2 and 3. This now makes for a more succinct manuscript with the focus on the key result.

*The Discussion also can be shortened and authors should stay focused on the novel and important findings. The novel finding is the predictive value of HIV DNA and time to viral rebound and keeping the focus on this will strengthen the manuscript. The finding is important as it can be used as a predictive marker in advancing the HIV cure research agenda. This finding and its significance is lost in the detailed presentation and discussion of novel and not so novel data*.

The Discussion has been shortened by removing all sections relating to immunological data. Whilst we feel these data are important, we acknowledge they might distract from the primary finding, and so have removed them completely. The manuscript, including the discussion, is now focused on HIV-1 DNA quantification and implications for control.

*Reviewers comments build around this novel finding for a more nuanced picture to emerge in terms of request for more details on measurement of DNA*, *for example an explanation on the somewhat paradoxical situation as to why free DNA is more important than integrated DNA…*

We acknowledge that our data raise some interesting questions. The greater impact of ‘Total’ vs ‘Integrated’ on our outcome measures is interesting, but we do not feel this is necessarily paradoxical. In purist terms the integrated DNA should reflect the true ‘reservoir’ of replication competent viral DNA, however the presence of unintegrated genomes (as reflected in the greater absolute quantity of Total DNA) may be indicative of slower dynamics of ART suppression, compartmentalization of replication or even on-going replication in the face of ART, depending on one’s reading of the literature. To fully understand these issues will require larger studies, as currently planned by the ACTG in the US, and ourselves in the UK. We have explored these questions in the discussion, but we feel that a detailed analysis of these variables would be better undertaken through these larger cohorts designed to look at these questions, and we hope the reviewers will be accepting of this approach.

*…and to the analysis of the DNA as a continuous variable rather than as a dichotomy*.

To be thorough, we analysed DNA as both a continuous and categorical variable in the Cox models and Kaplan–Meier analyses, respectively. The results from these two approaches were consistent and are presented in the paper and the supplementary material.

*A separate issue relates to new data on the Mississippi baby that was announced subsequent to the submission of this manuscript, namely the baby has detectable virus and this manuscript needs to be updated accordingly*.

This has now been done.

*Minor*
*comments:*

*The use of the term ‘reservoir’ when measuring HIV DNA will not be well-received by many in the field. This term should be used sparingly, or the authors should clarify what they mean by this term*.

We agree. We had tried to use the term sparingly; it is mostly found in the Introduction section, but we have taken steps to limit its use further. However, as it is part of the lexicon of HIV Cure research we have also defined in the text what the word ‘reservoir’ refers to in this manuscript, to ensure that where it is used the reader will understand the context.I.e ‘the population of HIV-1-infected cells that persist during ART and which are the source of rebound viraemia on stopping therapy’.

*In the Introduction, the authors suggest that the VISCONTI cohort argue against using HIV DNA as a predictor of what happens in absence of therapy. This may be true when compared to possible cure cases, but not when compared to the general HIV-infected population. The VISCONTI cohort is notable for having had a very low amount of HIV DNA, comparable to ‘elite’ controllers. The French team has argued that low DNA contributed to the outcome in this cohort. This text might be modified accordingly*.

We have re-written this section at the end of the Introduction, and also now include a comment on the Mississippi baby. The point being made is that with the few cases studies available HIV-1 DNA has not proved absolutely predictive (no DNA associated with rebound and cure in the Boston and Berlin transplants, respectively, and small amounts of DNA associated with on-going cure and rebound in the VISCONTIs and Mississippi case, respectively), and hence we need these larger cohort analyses.

*The inverse correlation between* CD4*+ T cell count and frequency HIV DNA in* CD4*+ T cells suggests that the total burden of HIV (per unit of blood) may be relatively stable. This brings up a constant dilemma with these cell-based assays: should the frequency of infection or total amount of infection be used in the analyses. Assessing whether the amount of HIV DNA in blood rather than frequency of HIV DNA in* CD4*+ T cells might provide unique insights, and might be considered for this paper, or future analyses*.

We agree this is an important point: certainly there is a lack of consistency in how DNA levels are reported across the literature, e.g. PBMCs vs CD4s, per million cells vs per ug DNA vs per blood volume. This does need to be sorted out and may also have mechanistic implications; however we would argue that this manuscript is not the place. The reviewers offer the option of ‘future analyses’ and we agree this is the appropriate route in this case. We are working on a broader analysis of the interpretation of different biomarkers and viral rebound (presented in part as a poster at CROI 2014), and we feel this analysis would be much better placed in this context.

*The association of HLA and immune responses on HIV replication has been the focus of decades worth of research. Given the strong association between HIV RNA and HIV DNA in untreated state, it is not surprising that immune response might be negatively correlated with HIV DNA during pre-therapy state. The functional assay used (ELISpot interferon-gamma) is less robust and informative of most assays now being applied to these types of studies. The data as presented are fine, but it makes it more challenging for a reader to find the more innovative data*.

A number of the authors on this manuscript have been heavily invested in understanding the HLA Class I-restricted immune response in HIV infection (Phillips, Kelleher, Frater, Carrington) over many years, hence the interest in this paper. Whereas we do not agree with the sentiment that ELISpot is less robust and informative than other assays (despite significant technological advances, ELISpot data remains as informative as other approaches in clinical cohorts), we acknowledge the inclusion of these data might confuse the overall message. We have therefore removed the immunological sections in their entirety.

*Presumably the association between immune response (HLA, ELIspot) and HIV DNA was similar to that observed between immune response and HIV RNA levels. This might be tested. Again, it is unclear if there is really much new in this analysis. The fact that HIV DNA predicted disease progression independent of HIV RNA is novel, and supports doing a similar analysis with the immunologic and genetic data*.

See section above. All functional immunology has been removed to make for a clearer story.

*The paper is generally easy to follow. The last sentence in the Results is not clear, however. The authors might wish to restate these findings*.

This has been done: ‘In summary, we find evidence that HIV-1 DNA is a significant predictor of the duration of viral remission and magnitude of the initial rebound following TI. This, if confirmed in larger studies, would have implications for those designing protocols for ART-reintroduction following viral rebound in TI studies.’

*The paper is too long. The Discussion could be cut*.

The paper has been cut by over 1000 words, including the Discussion.

Figure 5
*would be clearer if it had markers that showed clearly when ART was started and interrupted*.

The figure now includes a shaded region to show the period when ART is given.